# International Comparison of the Levels and Potential Correlates of Objectively Measured Sedentary Time and Physical Activity among Three-to-Four-Year-Old Children

**DOI:** 10.3390/ijerph16111929

**Published:** 2019-05-31

**Authors:** Kaiseree I Dias, James White, Russell Jago, Greet Cardon, Rachel Davey, Kathleen F Janz, Russell R Pate, Jardena J Puder, John J Reilly, Ruth Kipping

**Affiliations:** 1Population Health Sciences, Bristol Medical School, University of Bristol, Oakfield House, Oakfield Grove, Clifton, Bristol BS8 2BN, UK; ruth.kipping@bristol.ac.uk; 2Centre for Trials Research, School of Medicine, Cardiff University, 4th Floor Neuadd Meirionnydd, Heath Park, Cardiff CF14 4YS, UK; whitej11@cardiff.ac.uk; 3Centre for Exercise, Nutrition and Health Sciences, School for Policy Studies, University of Bristol, 8 Priory Road, Bristol BS8 1TZ, UK; russ.jago@bristol.ac.uk; 4Department of Movement and Sports Sciences, Ghent University, 9000 Ghent, Belgium; Greet.Cardon@UGent.be; 5Centre for Research & Action in Public Health, Health Research Institute, University of Canberra, Canberra, ACT 2601, Australia; rachel.davey@canberra.edu.au; 6Department of Health and Human Physiology, University of Iowa, Iowa City, IA 52242, USA; kathleen-janz@uiowa.edu; 7Department of Exercise Science, University of South Carolina, Columbia, SC 29208, USA; RPATE@mailbox.sc.edu; 8Obstetric service, Department Woman-Mother-Child, Lausanne University Hospital, 1005 Lausanne, Switzerland; jardena.puder@chuv.ch; 9School of Psychological Sciences and Health, University of Strathclyde, Glasgow G1 1QE, UK; john.j.reilly@strath.ac.uk

**Keywords:** child, preschool, accelerometry, physical activity, sedentary behavior, cross-sectional studies, ICAD

## Abstract

Physical activity (PA) patterns track from childhood through to adulthood. The study aimed to determine the levels and correlates of sedentary time (ST), total PA (TPA), and moderate-to-vigorous PA (MVPA) in preschool-aged children. We conducted cross-sectional analyses of 1052 children aged three-to-four-years-old from six studies included in the International Children’s Accelerometry Database. Multilevel linear regression models adjusting for age, gender, season, minutes of wear time, and study clustering effects were used to estimate associations between age, gender, country, season, ethnicity, parental education, day of the week, time of sunrise, time of sunset, and hours of daylight and the daily minutes spent in ST, TPA, and MVPA. Across the UK, Switzerland, Belgium, and the USA, children in our analysis sample spent 490 min in ST per day and 30.0% and 21.2% of children did not engage in recommended daily TPA (≥180 min) and MVPA (≥60 min) guidelines. There was evidence for an association between all 10 potential correlates analyzed and at least one of the outcome variables; average daily minutes spent in ST, TPA and/or MVPA. These correlates can inform the design of public health interventions internationally to decrease ST and increase PA in preschoolers.

## 1. Introduction

Physical activity (PA) patterns track from childhood through to adulthood [1], making preschool-aged children an important population to target for physical activity interventions. Being physically active during the early years is associated with improved adiposity, cardiometabolic health indicators, motor skill development, bone and skeletal health, cognitive development, and psychosocial health [2]. Current Canadian [3] and Australian [4] guidelines advise that children aged two-to-five-years-old should not be sedentary for periods of over 60 min at a time. The Canadian, Australian, USA [5], and UK [6] guidelines also specify that children under the age of five, who can walk unaided, should be physically active for at least 180 min per day and should spend at least 60 min of this time in moderate-to-vigorous PA (MVPA) [3,4]. Only a few studies have looked at the proportion of preschool-aged children meeting these PA guidelines using objective measures of PA. Two studies from the UK [7,8] found that 100% of children aged three-to-four-years met the recommended ≥180 min a day of total PA (TPA) whereas a Belgian [9], Australian [10], and Canadian [11] study found that 11.0%, 5.1%, and 83.8% of preschool-aged children met these guidelines, respectively. Furthermore, the Canadian study found that 13.7% of five-year-olds spent ≥60 min in MVPA per day [11]. It is, however, not possible to establish whether children in the UK are more physically active than children in Canada and Australia due to differences in definitions of accelerometer wear time applied across these studies [12,13]. This emphasizes the importance of applying standardized data processing methods to ensure comparisons across countries are valid.

A key stage in the development of behavior change interventions is identifying variables which could either be potential targets to change behavior (mediators) or variables that could affect the outcome of the behavior change program (moderators). Therefore, identifying the key correlates of preschoolers’ sedentary time (ST) and PA is important for designing effective behavior change programs [14]. Narrative reviews have assessed the correlates of ST [15,16] and PA [16,17] in preschool-aged children. Across the reviews, there was inconsistent support for associations between ST and PA with correlates. For example, one review concluded boys were more active than girls [17], whereas the other did not [16]. Conflicting findings were also observed for day of the week, where one review found no association [17] and the other found a positive association [16] with physical activity. Both found no association between age, ethnicity, season, or parent education with physical activity [16,17]. All three reviews were limited in that they included studies which used self-report measures of ST and PA, which may not accurately detect associations. Since these reviews were published, there have been a few additional studies which have assessed the correlates of ST and/or PA in preschool-aged children using objectively measured accelerometry data [18,19,20,21,22,23,24,25]. The findings across these studies are similarly inconsistent, and there is the issue of comparability as they have processed accelerometry data using different methods of processing and analyzing data [18,19,20,21,22,23,24,25]. In addition, none of the studies have made cross-national comparisons of the proportion of children meeting guideline levels of ST, TPA, and MVPA and using a standardized method of processing accelerometry data. Accordingly, we aim to determine the levels and correlates of ST, TPA, and MVPA in children aged three-to-four-years-old using data from four countries, applying a consistent approach to data processing.

## 2. Methods

### 2.1. Study Design

Cross-sectional analyses were carried out on data obtained from the International Children’s Accelerometry Database (ICAD), which has been described in detail elsewhere [26]. In brief, the ICAD is a pooled database of raw Actigraph accelerometer (Actigraph LLC, Pensacola, FL, USA) data files and accompanying demographic, anthropometric, and health data collected from children (2–18-years) between 1997 and 2009. Data were pooled from 20 studies conducted in 11 countries and included cross-sectional, longitudinal, intervention, and closed cohort studies. Data were reduced using standardized techniques to allow for comparison of physical activity outcome variables across studies (see below). Formal data sharing agreements were established between all study authors and ICAD. All studies consulted their individual research boards to confirm that appropriate ethical approval had been attained for contributing data.

### 2.2. Participants

For this paper, the analytical sample consists of children aged three-to-four-years-old who had at least three days (week and/or weekend days) of valid accelerometry data [27]. To maintain the independence of the observations, follow-up waves of data were excluded from the analyses (*n =* 17). Participants aged two-years-old (*n* = 17) and participants from Australia (*n =* 7) were excluded due to the very small sample sizes for these groups. Data for the analysis sample were extracted from six studies: Ballabeina Study [28]—Switzerland; Belgium Pre-School Study [29,30,31]—Belgium; Children’s Health and Activity Monitoring Programme (CHAMPS) UK [32,33]—UK; CHAMPS U.S. [34]—USA.; Iowa Bone Development Study (IBDS) [35,36]—USA.; and Movement and Activity Glasgow Intervention in Children (MAGIC) [37]—UK.

### 2.3. Physical Activity Measurement

Physical activity was measured using waist-worn, uniaxial Actigraph accelerometers (models 7164, 71256, and GT1M). Raw data files were processed using KineSoft version 3.3.20 (KineSoft, Sakatchewan, SK, Canada). Non-wear time was defined as periods of 60 min of consecutive zeros allowing for two minutes of non-zero interruptions [38]. A day was considered valid if there was at least 480 min of accelerometry data [27]. The analysis of physical activity data was restricted to 06:00 and 21:59 to exclude the times when the children would be asleep [39]. When looking at physical activity patterns across the day, hours with less than two minutes of wear time were removed from the analyses [38]. Physical activity thresholds available in the ICAD were those specified by Puyau et al. [40] and Pate et al. [41]: sedentary (<800 cpm) [40], TPA (≥800 cpm) [40] and MVPA (≥1680 cpm) [41]. Mean hourly, daily, weekday, and weekend minutes spent in ST, TPA, and MVPA across the whole week were the outcome variables.

### 2.4. Variables

The following 10 potential correlates were examined: age, gender, country, season, ethnicity, parental education, day of the week (weekday vs weekend), time of sunrise, time of sunset, and hours of daylight. These variables were explored based on a combination of what was available and what had been identified as potential correlates by previous studies [15,16,17]. Ethnicity data were available from three studies in our analysis sample and categorized as white or other (non-white). Parental education was available from four studies and was dichotomized into ‘up to and including completion of compulsory education including vocational training’ and ‘any post-compulsory education including vocational training’ as a measure of socioeconomic status. The season, time of sunrise, time of sunset, and hours of daylight variables were derived from the date that the accelerometer started collecting data and the city, or nearest city, where the study took place using the website www.timeanddate.com. The countries in our sample were all in the Northern Hemisphere and therefore, had the same seasons (spring: March–May; summer: June–August; autumn: September–November; winter: December–February). The time of sunrise variable, time of sunset, and hours of daylight variables were categorized into before and after 07:00; before and after 19:00; and less than or more than 12 h, respectively.

### 2.5. Statistical Analyses

Participant characteristics were summarized using frequencies and percentages for categorical data. The percentage of children meeting the recommended daily guidelines of ≥180 min of TPA [3,4,5,6] and ≥60 min MVPA [3,4] were compared across categories of each correlate using chi-squared tests. Mean minutes spent in ST, TPA, and MVPA were plotted for every hour between 06:00 and 21:59. Adjusted multilevel linear regression models were used to determine the association between ST, TPA, and MVPA for each potential correlate. Models were adjusted for age, gender, season, minutes of wear time, and study clustering effects. Linear regression analyses were undertaken assuming a linear relationship, multivariate normality, homoscedasticity, and little multicollinearity which were tested via inspection of scatter plots of the outcomes vs. the independent variable; histograms of the outcome variables; scatter plots of the residual errors vs. the linear predictor; and variance inflation factors of the variables included in the models, respectively. Results from the assumption tests clarified that these assumptions had been met (data not shown). Intraclass correlation coefficients (ICCs) and R-squared values (R^2^), as proposed by Snijders and Bosker [42], were calculated for each of the models. Some studies [43,44] suggest that four valid days of accelerometry data are needed to reliably measure ST/PA levels to achieve an ICC of ≥0.7 when using the accelerometry thresholds specified in our analyses [40,41]. Sensitivity analyses were, therefore, carried out on a sample where participants had at least four days of valid accelerometry data. All analyses were carried out in Stata v15 (StataCorp, College Station, TX, USA).

## 3. Results

### 3.1. Participant Characteristics

The 1052 participants in the analysis sample (Table 1) contributed an average of 4.82 days of data comprising 3.79 weekdays and 1.03 weekend days. The average daily wear time between 6:00 to 21:59 was 697.27 min (see Appendix A). Data were collected between September 1998 and June 2009. Out of the six studies which contributed data, most participants were from the UK-based MAGIC study (36.8%).

### 3.2. Percentage of Children Meeting Canadian, Australian, USA and UK Guidelines for Sedentary Time, Total Physical Activity, and Moderate-to-Vigorous Physical Activity

Participants spent an average of 490.18 min per day in ST (see Appendix A). Table 2 shows that 70.0% of participants met recommended daily guidelines of ≥180 min of TPA and 78.8% of participants met daily guidelines of ≥60 min of MVPA, based on TPA (≥800 cpm) [40] and MVPA (≥1680 cpm) [41] thresholds specified by Puyau et al. and Pate et al. A greater percentage of four-year -olds than three-year-olds and boys than girls met the recommended guidelines for TPA and MVPA. Our findings suggest that the percentage of children reaching TPA and MVPA guidelines varied between the different countries. The lowest percentage of children achieving guideline MVPA levels was observed in Belgium (50.0%), and the highest percentage was observed in the USA (88.7%). The percentage of children reaching the recommended TPA and MVPA levels increased from winter through to summer before it decreased in autumn and was greater on weekdays compared to weekends. A greater percentage of non-white children met the MVPA guidelines compared to white children (92.7% vs. 78.0%, *X*^2^ = 18.40, *p* < 0.001). When the hours of daylight were more than 12 h, a greater percentage of children met TPA (76.2% vs. 65.0%, *X*^2^ = 15.52, *p* < 0.001) and MVPA (85.1% vs. 73.9%, *X*^2^ = 19.62, *p* < 0.001) guidelines compared to when days were less than 12 h long. A greater percentage of children met TPA and MVPA guidelines when the time of sunrise was before 07:00 and the time of sunset was after 19:00 compared to being after 07:00 and before 19:00. No differences were observed for parental education.

### 3.3. Patterns of Sedentary Time and Physical Activity across the Day

Figure 1 shows that ST levels increase until around 09:00 and then decrease throughout the day, whereas TPA and MVPA levels increase throughout the day with variations in PA by country, day of the week and hours of daylight between 11:00 and 15:00. Figure 1A suggests that children from the USA showed a greater dip in TPA and MVPA levels between 11:00 and 15:00 than that observed around 12:00 in the UK and in Switzerland. Minutes spent in ST appear to have been higher on weekdays compared to weekends (Figure 1B) until 14:00 and 15:00 when minutes spent in ST became similar. On weekdays, the minutes spent in TPA and MVPA rose until 10:00 to 11:00 before dipping, whereas, on weekends, the minutes spent in TPA and MVPA increased gradually throughout the day before reaching a peak at the same time as weekdays at around 16:00. Between the hours 09:00 and 18:00, the minutes spent in ST were higher when the hours of daylight were less than 12 h compared to being more than 12 h (Figure 1C). The minutes spent in TPA and MVPA were noticeably higher when the hours of daylight were more than 12 h long, apart from the period before 09:00 and at the dipped levels observed between 12:00 and 15:00 where levels were similar to those observed on days which are less than 12 h long.

### 3.4. Correlates of Sedentary Time and Physical Activity in Preschool-Aged Children

Table 3 shows the adjusted associations between the potential correlates and average daily minutes spent in ST, TPA, and MVPA after adjusting for age, gender, season, minutes of wear time, and study level clustering. Minutes spent in ST were higher, while minutes spent in TPA were lower, in girls, winter, and children whose parental education levels were higher compared to boys, spring, and lower parental education levels, respectively. Children spent more minutes in ST on weekdays compared to weekends and in MVPA summer compared to winter. There was evidence that four-year-olds, boys, and non-white children spent more time in MVPA compared to three-year-olds, girls, and white children, respectively. Minutes spent in ST were lower and the time spent in TPA/MVPA was higher when the hours of daylight were greater, i.e., when the time of sunrise was before 07:00, time of sunset was after 19:00 and when the hours of daylight were longer than 12 h long. UK-based children spent more time in TPA and fewer minutes in ST compared to children from Switzerland, Belgium, and the USA but only spent more time in MVPA compared to Switzerland and Belgium. The unadjusted analysis findings are available in Appendix A.

## 4. Discussion

This study aimed to determine the levels and correlates of objectively measured ST, TPA, and MVPA in preschool-aged children using pooled data from the ICAD, which has been processed and analyzed using standardized methods. Across four high-income countries, three-to-four-year-olds were sedentary for an average of over 8 h per day. Thirty percent of the preschool-aged children were not engaging in the recommended ≥180 min of TPA, and 21.2% were not getting ≥60 min of MVPA per day. Data by each hour suggest that the minutes spent in ST decreased throughout the day, and the dips in TPA and MVPA levels generally observed between 11:00 and 15:00 were more prominent on weekdays compared to weekends, and in the USA compared to the other three countries. There was evidence for an association between all 10 potential correlates analyzed and at least one of the outcome variables; average daily minutes spent in ST, TPA, and/or MVPA.

Overall, 70.0% of our sample achieved ≥180 min of TPA, which differs to findings from the UK [7,8], Belgian [9], Australian [10], and Canadian [11] studies which found that 100%, 11.0%, 5.1%, and 83.8% of preschool-aged children achieved recommended guidelines, respectively. Compared to the TPA threshold used in our study (≥800 cpm) [40], the two UK studies and Canadian study used thresholds of ≥152 cpm [7,41], ≥20 cpm [8], and ≥100 cpm [11,45], respectively. These thresholds are lower than the ones used in this study, and therefore, a greater percentage of their participants could have achieved the PA guidelines. Similarly, the Belgian [9] and Australian [10] studies used thresholds described by Reilly et al. (≥1100 cpm) [46] and Sirard et al. (3-years: ≥1208 cpm; 4-years: ≥1456 cpm; 5-years: ≥1596 cpm) [47] which are higher than our study and may explain why such a small percentage of their samples achieved daily TPA guidelines compared to our sample. The Canadian study [11] found that 13.7% of five-year-olds spent ≥60 min in MVPA per day, whereas 78.8% of our sample achieved these recommendations. In comparing the different thresholds used in the studies, one might expect the percentage of our participants who achieved the recommended MVPA guidelines to be lower than the Canadian study, as they used a lower MVPA threshold, but this is not the case (78.8% vs. 13.7%). This highlights the difficulties with making comparisons between studies due to study differences in not only the accelerometry thresholds for different intensities but also the exclusion of participants based on insufficient accelerometry wear time [12,13]. As we used a pooled dataset in which data has been processed in the same way across studies [26], the differences we have found between countries cannot be attributed to differences in data processing. We found that the greatest proportion of children reaching recommended TPA and MVPA guidelines were in the USA, followed by the UK, Switzerland, and Belgium. An exploratory subgroup analysis (data not shown), found the percentage of four-year-olds was highest in Switzerland followed by the UK, Belgium, and the USA, and the ratio of girls to boys was similar across the four countries. Most of the data were collected in autumn for UK, USA, and Switzerland-based children and in spring for Belgium-based children. Minutes of wear time were highest in the USA followed by Switzerland, Belgium, and the UK (see Appendix A). It is, therefore, unlikely that the between-country differences are a result of age, gender, season or minutes of wear time differences; which had been adjusted for in the regression analyses.

Visual inspection of the plots of ST by hour suggested that children spent fewer minutes in ST as the day progressed after an initial increase in ST levels until 09:00. In general, the figures showed that TPA and MVPA levels peaked either side of 11:00 and 15:00, with the peak observed after 15:00 being the highest level of PA reached in the day. Our findings are comparable to results from an Australian-based study [48] which found that ST was at its lowest and MVPA levels were at their highest from the mid-afternoon through to the evening on both weekdays and weekends. The dip in TPA and MVPA levels observed in the USA between 11:00 and 15:00 was greater in width and magnitude than the dips observed in Switzerland and the UK at around 12:00, which may represent differences in the childcare routines practiced by the different countries. The patterns of ST and PA in Belgium throughout the day were harder to distinguish, which is likely due to the lower sample size which contributed data. The dips in TPA and MVPA levels were more prominent on weekdays compared to weekends from 11:00 to 15:00 which may be representative of preschool lunchtime and napping procedures; however, we do not have preschool attendance data available to draw such conclusions. Reports from international early years settings suggest that compulsory sleep times are commonplace in childcare settings [49,50,51] which highlights the importance of having this information on policies and practices on sleep times available for analysis. The two peaks of MVPA levels in the morning and evening are elevated when the days are longer than 12 h long compared to being less than 12 h long. It could be suggested that more opportunities for outdoor play are available for children when the days are lighter, which is contributing to these higher activity levels [16].

Our findings from the regression analyses replicate those from other studies which found no association between age and ST [18,19,20,21,52,53,54] and found that girls were more sedentary than boys [18,19,22,52]. Children in Switzerland, Belgium, and the USA spent more minutes in ST than children in the UK. Our findings replicate results from another study which found that children were more sedentary in winter [22] compared to spring, whereas other studies only found that they were less sedentary in autumn [54] compared to spring or did not find an association between seasonality and ST [21,53]. We found no association between ST and ethnicity which is consistent with another study [19], and we found a positive association between ST and parental education which is not consistent with other studies [18,19,22,23,24,53] that found no association with ST. It is possible that this is a chance finding due to the smaller sample sizes of participants who had ethnicity (*n* = 419) and parental education (*n* = 386) data. We found that children were more sedentary on weekdays compared to weekends, which is consistent with a previous study which found that hour-by-hour ST levels tended to be lower on weekends compared to weekdays [48].

It is well established that older preschoolers are more active than younger preschoolers [18,21,22,25,54] and that boys are more active than girls [18,21,22,25,39,53,54,55,56,57] although we did not observe a difference in minutes spent in TPA between three- and four-year-olds. We found that children in the UK were more physically active than children in Switzerland, Belgium, and the U.S.A, but there was less evidence to show that MVPA levels were higher in the USA. Similar to our findings, it has been observed elsewhere that children spend more time in MVPA in summer compared to winter [53] and another study observed that children were only more active in spring and not in summer compared to winter [39], although this was for MVPA not TPA as in our study. Another study found that children spent more time in TPA in summer compared to other seasons, [21] whereas others found that children spent more time in MVPA in summer and less time in winter compared to spring [22]. Previous studies found no associations with ethnicity [17] or parent education [22,23,25,53,56,58] and PA measures, whereas we found that non-white children spent more time in MVPA than white children, and children whose parents had lower education levels spent greater time in TPA than those with higher parental education levels. We did not observe a difference between weekday and weekend data PA levels which is consistent with one study [59], whereas another study found that children spent more time in MVPA on weekends compared to weekdays [48]. When the hours of daylight were longer (including an earlier sunrise and later sunset) the children spent a greater time in PA and fewer minutes in ST which is comparable to a study looking at older children (5–16-years) which found that longer evening sunlight was associated with increased daily physical activity [60].

The strength of this study is that it adds to the limited literature on levels and correlates of objectively measured ST and PA in preschool-aged children. There have been a particularly limited number of studies which have previously examined ST and PA by ethnicity and parental education variables. Estimates for the ethnicity and parental education variables had large amounts of missingness. Therefore, we have assumed that these estimates would apply if the data were not missing. To our knowledge, there have been no previous literatures looking at differences in objectively measured ST/PA by country, time of sunrise, time of sunset, and hours of daylight in this age group. Consequently, there were no previous references to base our daylight variable categorizations on which may be a limitation in the analyses. As the data from the different studies within the ICAD have been processed in the same way, the results we present are a ‘fair’ comparison of levels of ST/PA across different countries which have not previously been possible. It is important to acknowledge that there are a relatively small number of children in each of the countries that were included in the sample. Therefore, our findings are not representative of each country’s population. The studies included in the analysis sample are all based in high-income countries; therefore, our results may not be generalizable to low-to-middle-income countries. Data used in this study were collected between 1998 and 2009. Therefore, the results may not be generalizable to the current cohorts of preschool-aged children, especially given childrens’ changing access to screens [61]. We were not provided with the raw data, so it was not possible to accurately identify the number of times children exceeded being sedentary for periods ≥60 min at a time; therefore, we were not able to measure the proportion of children meeting recommended ST guidelines. Based on the information provided in the ICAD [26] codebook, there is no information on napping, and as such, it appears that nap/sleep time may have been considered as non-wear time which may have led to the overestimation of ST levels and the underestimation of PA levels. The data is compositional in nature, therefore using compositional data techniques as opposed to standard techniques may have produced different results [62]. We did not have data available about childcare differences within and between samples, which could have been used to interpret our findings and to potentially explain between-country differences. Data from longitudinal studies can estimate modifiable factors associated with changes in ST and PA [63], whereas our cross-sectional study is limited in only providing evidence of associations.

## 5. Conclusions

Using data from four high-income countries, we found that children spent over eight hours per day in ST and 30.0% and 21.2% of children were not engaging in recommended daily amount of TPA (≥180 min) and MVPA (≥60 min), respectively. The minutes spent in ST decreased throughout the day and the dips in TPA and MVPA levels observed between 11:00 and 15:00 were greater in the USA compared to Switzerland, Belgium, and the UK and on weekdays compared to weekends. Age, gender, country, season, ethnicity, parental education, day of the week, time of sunrise, time of sunset, and hours of daylight were all identified as potential correlates of minutes spent in ST and/or TPA and/or MVPA. The associations between ethnicity and parental education with ST and PA were derived from smaller sample sizes and should be investigated further in a larger population. Internationally, there is a need for public health interventions, to decrease ST and increase PA levels in three-to-four-year-olds. The potential correlates identified in this study can be considered in designing these public health interventions. However, further research is needed to determine modifiable factors associated with ST and PA in preschool-aged children to inform effective behavior change programs.

## Figures and Tables

**Figure 1 ijerph-16-01929-f001:**
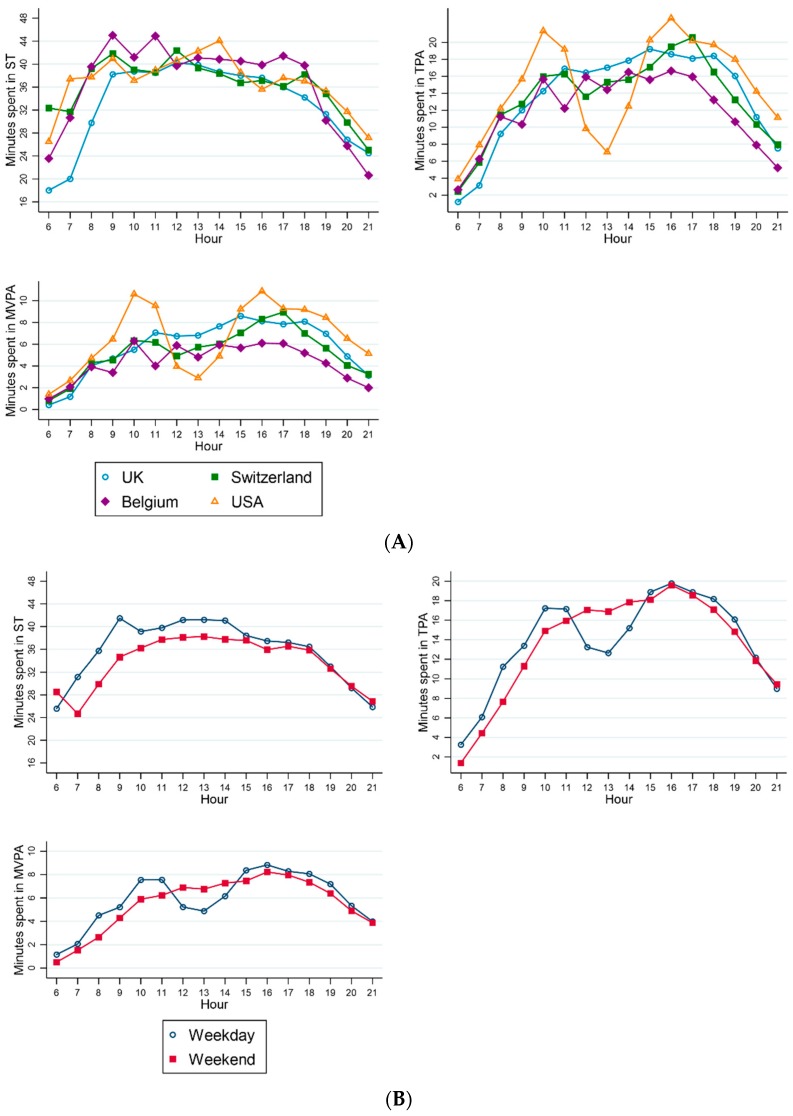
Mean minutes spent in sedentary time, total physical activity, and moderate-to-vigorous physical activity according to the hours of the day by (**A**) country, (**B**) day of the week (weekday vs. weekend), and (**C**) hours of daylight. Figure 1A shows the by country differences in minutes spent in ST, TPA, and MVPA by hour. (Figure 1B shows the differences in minutes spent in ST, TPA, and MVPA by hour on weekdays compared to weekends. Figure 1C shows the differences in minutes spent in ST, TPA, and MVPA by hour when the hours of daylight are less than 12 h long compared to being more than 12 h long.)

**Table 1 ijerph-16-01929-t001:** Sociodemographic characteristics of children.

Characteristic	*N* (%)
Overall	1052 (100.00)
Age	
3	343 (32.60)
4	709 (67.40)
Gender	
Male	528 (50.19)
Female	524 (49.81)
Country	
UK	426 (40.49)
Switzerland	142 (13.50)
Belgium	104 (9.89)
USA	380 (36.12)
Season	
Winter	136 (12.93)
Spring	110 (10.46)
Summer	117 (11.12)
Autumn	689 (65.49)
Ethnicity	
White	200 (19.01)
Other	219 (20.82)
Missing/Not available	633 (60.17)
Parental Education	
Up to and including completion of compulsory vocational training	86 (8.17)
Any post-compulsory education including vocational training	300 (28.52)
Missing/Not available	666 (63.31)
Day of the Week	
Weekday	1052 (100.00)
Weekend	626 (59.51)
Time of Sunrise	
Before 07:00	433 (41.16)
After 07:00	619 (58.84)
Time of Sunset	
Before 19:00	548 (52.09)
After 19:00	504 (47.91)
Hours of Daylight	
Less than 12 h	589 (55.99)
More than 12 h	463 (44.01)
Study	
Ballabeina	142 (13.50)
Belgium Pre-School	104 (9.89)
CHAMPS U.S.	361 (34.32)
MAGIC	387 (36.79)
CHAMPS UK	39 (3.71)
IBDS	19 (1.81)

**Table 2 ijerph-16-01929-t002:** Frequency and percentage of children meeting internationally recognized guidelines of ≥180 min of total physical activity per day and ≥60 min of moderate-to-vigorous physical activity per day by the different correlates.

Correlate	N	≥180 min of TPA	*X* ^2^	*p*	≥60 min of MVPA	*X* ^2^	*p*
Overall	1052	736 (69.96)	N/A	N/A	829 (78.80)	N/A	N/A
Age							
3	343	223 (65.01)			257 (74.93)		
4	709	513 (72.36)	5.93	0.015	572 (80.68)	4.58	0.032
Gender							
Male	528	406 (76.89)			451 (85.42)		
Female	524	330 (62.98)	24.24	<0.001	378 (72.14)	27.76	<0.001
Country							
UK	426	297 (69.72)			332 (77.93)		
Switzerland	142	99 (69.72)			108 (76.06)		
Belgium	104	46 (44.23)			52 (50.00)		
USA	380	294 (77.37)	42.70	<0.001	337 (88.68)	74.70	<0.001
Season							
Winter	136	82 (60.29)			93 (68.38)		
Spring	110	79 (71.82)			85 (77.27)		
Summer	117	90 (76.92)			106 (90.60)		
Autumn	689	485 (70.39)	8.99	0.029	545 (79.10)	18.78	<0.001
Ethnicity							
White	200	143 (71.50)			156 (78.00)		
Other	219	171 (78.08)	2.41	0.120	203 (92.69)	18.40	<0.001
Parental Education							
Up to and including completion of compulsory vocational training	86	73 (84.88)			81 (94.19)		
Any post-compulsory education including vocational training	300	226 (75.33)	3.49	0.062	260 (86.67)	3.67	0.055
Weekday vs. Weekend							
Weekday	1052	720 (68.44)			813 (77.28)		
Weekend	626	386 (61.66)	8.03	0.005	423 (67.57)	19.07	<0.001
Time of Sunrise							
Before 07:00	433	344 (79.45)			382 (88.22)		
After 07:00	619	392 (63.33)	31.49	<0.001	447 (72.21)	39.09	<0.001
Time of Sunset							
Before 19:00	548	350 (63.87)			399 (72.81)		
After 19:00	504	386 (76.59)	20.21	<0.001	430 (85.32)	24.59	<0.001
Hours of daylight							
Less than 12 h	589	383 (65.03)			435 (73.85)		
More than 12 h	463	353 (76.24)	15.52	<0.001	394 (85.10)	19.62	<0.001

**Table 3 ijerph-16-01929-t003:** Multi-level adjusted associations between potential correlates and average daily minutes spent in sedentary time, total physical activity, and moderate-to-vigorous physical activity in children aged 3-to-4-years-old.

**Sedentary Time**						
**Correlate (Reference Category)**	**N**	**β**	**(95% CI)**	***p***	**ICC**	**R^2^**
Age (ref = 3−years)	1052	−3.54	(−9.85, 2.77)	0.272	0.085	0.635
Gender (ref = Male)	1052	17.81	(12.14, 23.49)	<0.001	0.085	0.635
Country (ref = UK)	1052				0.000	0.944
Switzerland		22.06	(12.09, 32.03)	<0.001		
Belgium		36.68	(25.34, 48.02)	<0.001		
USA		10.73	(2.54, 18.91)	0.010		
Season (ref = Winter)	1052				0.085	0.635
Spring		−14.01	(−26.28, −1.74)	0.025		
Summer		−12.16	(−24.90, 0.57)	0.061		
Autumn		0.93	(−9.42, 11.28)	0.861		
Ethnicity (ref = White)	419	−3.07	(−12.71, 6.56)	0.532	0.000	0.903
Parental Education (ref = Up to/including compulsory education)	386	14.91	(3.65, 26.17)	0.009	0.000	0.609
Weekday vs. Weekend (ref = Weekday)	1678	−33.60	(−40.03, −27.18)	<0.001	0.084	0.511
Time of Sunrise (ref = Before 07:00)	1052	10.80	(3.88, 17.72)	0.002	0.070	0.696
Time of Sunset (ref = Before 19:00)	1052	−15.20	(−22.20, −8.19)	<0.001	0.089	0.626
Hours of daylight (ref = Less than 12 h)	1052	−10.33	(−17.53, −3.13)	0.005	0.085	0.636
**Total Physical Activity**						
**Correlate (Reference Category)**	**N**	**β**	**(95% CI)**	***p***	**ICC**	**R^2^**
Age (ref = 3−years)	1052	3.54	(−2.77, 9.85)	0.272	0.085	0.273
Gender (ref = Male)	1052	−17.81	(−23.48, −12.14)	<0.001	0.085	0.273
Country (ref = UK)	1052				0.000	0.888
Switzerland		−22.05	(−32.02, −12.08)	<0.001		
Belgium		−36.68	(−48.02, −25.35)	<0.001		
USA		−10.72	(−18.90, −2.53)	0.010		
Season (ref = Winter)	1052				0.085	0.273
Spring		14.00	(1.73, 26.28)	0.025		
Summer		12.16	(−0.58, 24.89)	0.061		
Autumn		−0.93	(−11.28, 9.42)	0.860		
Ethnicity (ref = White)	419	3.07	(−6.56, 12.71)	0.532	0.000	0.884
Parental Education (ref = Up to/including compulsory education)	386	−14.91	(−26.17, −3.65)	0.009	0.000	0.203
Weekday vs Weekend (ref = Weekday)	1678	−3.65	(−9.30, 2.00)	0.205	0.096	0.224
Time of Sunrise (ref = Before 07:00)	1052	−10.80	(−17.72, −3.87)	0.002	0.070	0.395
Time of Sunset (ref = Before 19:00)	1052	15.20	(8.19, 22.20)	<0.001	0.089	0.256
Hours of daylight (ref = Less than 12 h)	1052	10.33	(3.12, 17.53)	0.005	0.085	0.276
**Moderate−to−Vigorous Physical Activity**						
**Correlate (Reference Category)**	**N**	**β**	**(95% CI)**	***p***	**ICC**	**R^2^**
Age (ref = 3−years)	1052	4.91	(0.77, 9.05)	0.020	0.095	0.299
Gender (ref = Male)	1052	−14.94	(−18.66, −11.21)	<0.001	0.095	0.299
Country (ref = UK)	1052				0.000	0.904
Switzerland		−15.93	(−22.46, −9.41)	<0.001		
Belgium		−22.48	(−29.90, −15.05)	<0.001		
USA		4.06	(−1.30, 9.42)	0.137		
Season (ref = Winter)	1052				0.095	0.299
Spring		7.96	(−0.10, 16.03)	0.053		
Summer		11.94	(3.57, 20.32)	0.005		
Autumn		3.58	(−3.24, 10.39)	0.304		
Ethnicity (ref = White)	419	9.53	(2.89, 16.18)	0.005	0.000	0.865
Parental Education (ref = Up to/including compulsory education)	386	−7.75	(−15.59, 0.09)	0.053	0.000	0.149
Weekday vs Weekend (ref = Weekday)	1678	−1.39	(−4.96, 2.18)	0.446	0.095	0.289
Time of Sunrise (ref = Before 07:00)	1052	−4.96	(−9.52, −0.40)	0.033	0.086	0.364
Time of Sunset (ref = Before 19:00)	1052	9.47	(4.86, 14.08)	<0.001	0.099	0.281
Hours of daylight (ref = Less than 12 h)	1052	7.04	(2.30, 11.77)	0.004	0.098	0.284

Note: CI: Confidence Interval, ICC: Intraclass Correlation Coefficient. All models are adjusted for age, gender, season, minutes of wear time, and study clustering effects.

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
