# Peer review of "International Comparison of the Levels and Potential Correlates of Objectively Measured Sedentary Time and Physical Activity among Three-to-Four-Year-Old Children"

_ijerph, 2019, doi:10.3390/ijerph16111929_

Round 1
Reviewer 1 Report
This paper aims to inform the design of physical activity interventions in 3-4 year-old children in its explorations of correlates of the same. A major strength of this paper is that all physical activity data sets were derived from a unified device protocol and procedural checklist. At the same time, a few points of inquiry arise, as noted:
Lines 110: Please note if a weekend day was required for inclusion within the final analytic sample.
Line 134-136: The dichotomized times of before/after 07:00 and 19:00 for sunrise and sunset, respectively, may require additional discussion. Why were these particular cut points used to mark these respective times of the day?
Line 142: ‘were’ instead of ‘was’
Line 140-150: Please include a note about the descriptive statistics that were run. Additionally, please include a statement on the assumptions of the tests and whether they were found to be tenable in the specified models. Additionally, please include the ICC for the random intercept only model. How did the model account for the high degree of missingness in some variables that were included as covariates such as race/ethnicity and parental education; what was the impact of the missingness on the final analytic sample included in the models? Please include some discussion on the compositional nature of the data given that both ST and TPA are analyzed, but in separate models.
Gupta, N., Mathiassen, S. E., Mateu-Figueras, G., Heiden, M., Hallman, D. M., Jorgensen, M. B., & Holtermann, A. (2018). A comparison of standard and compositional data analysis in studies addressing group differences in sedentary behavior and physical activity. The International Journal of Behavioral Nutrition and Physical Activity, 15(1), 53. https://doi.org/10.1186/s12966-018-0685-1
Line 162-177: Please include effect sizes for chi-squared tests
Line 167-169: Are these statements qualitative comparisons, or were statistical tests run? Please clarify the statistical significance of the observed differences…
Table 2: Please include effect sizes
Figure 1: Please consider presenting these figures as % of wear time on the y-axis to help clarify, especially, the relationship between ST and TPA. Also, a footer that describes each of the subplots will help to reinforce the meaning of each aspect of the multipaneled figure. ST is used throughout the text referring to sedentary time; however, the figures use SB
Line 183-185: Instead of only (B) and (C) to identify subplots in Figure 1, please consider something like Figure 1B and Figure 1C to more easily identify what is meant by the letters main text body.
Lines 205- 217: What is the Adjusted R squared value for each of the respective models? Additionally, some of the covariates (e.g., season, time of sunrise, time of sunset, hours of daylight) seem like they might be highly correlated (i.e., may have shared variances) was there a search for optimal model fit when including these covariates? What were the considerations for multicollinearity between covariates in specifying the final model?
Line 258: ‘were’ instead of ‘was’
Lines 263-281: It remains unclear what statistical analyses were conducted to arrive at these discussion points, or if this is based upon qualitative description of the plots alone. Please clarify this point. It is stated that ST decreases, but the plots show that first ST increases, then it decreases after reaching a fairly clear peak point or else reaching several local maxima throughout the day...
Lines 329-331: Did you mean to say something about “sleep/nap” time here instead of “non-wear time”?
Author Response
17th May 2019
Dear Sir/Madam,
We appreciate the opportunity to resubmit our manuscript “International comparison of the levels and potential correlates of objectively measured sedentary time and physical activity among 3-4-year old children”, ID: ijerph-496064, for publication as an original article in the International Journal of Environmental Research and Public Health.
The comments from the reviewers helped us improve our manuscript. Our response to each comment is indicated in red text in the following pages. Revisions in the manuscript are shown as tracked changes. We hope that the responses and revisions are sufficient to make our manuscript suitable for publication in your journal.
We look forward to hearing from you at your earliest convenience.
Yours sincerely,
The study authors
Response to Reviewer 1 Comments
REVIEWER 1.1. Lines 110: Please note if a weekend day was required for inclusion within the final analytic sample.
RESPONSE 1.1. A weekend day was not required for inclusion within the final analytical sample which is consistent with the validation study that is referenced in the manuscript (see below). This has been noted in Line 102 to make this more explicit: “For this paper, the analytical sample consists of children aged three-to-four-years old who had at least three days (week and/or weekend days) of valid accelerometry data[27]”.
Hinkley, T.; O'Connell, E.; Okely, A.D.; Crawford, D.; Hesketh, K.; Salmon, J. Assessing volume of accelerometry data for reliability in preschool children. Med Sci Sports Exerc 2012, 44, 2436-2441, doi:https://dx.doi.org/10.1249/MSS.0b013e3182661478.
REVIEWER 1.2. Line 134-136: The dichotomized times of before/after 07:00 and 19:00 for sunrise and sunset, respectively, may require additional discussion. Why were these particular cut points used to mark these respective times of the day?
RESPONSE 1.2. As mentioned in Lines 342-344, there has been no previous literature looking at time of sunrise and time of sunset as potential correlates of preschool aged children’s ST/PA levels, for us to base our cut points on. Based on the data, we found that approximately 50% of the sample had data either side of these cut points which is why they were selected. We thank the reviewer for suggesting additional discussion and this has now been acknowledged as a limitation on Lines 344-345 to read “Consequently, there were no previous references to base our daylight variable categorizations on which may be a limitation in the analyses”.
REVIEWER 1.3. Line 142: ‘were’ instead of ‘was’
RESPONSE 1.3. This has been changed in the manuscript on Line 144.
REVIEWER 1.4. Line 140-150: Please include a note about the descriptive statistics that were run. Additionally, please include a statement on the assumptions of the tests and whether they were found to be tenable in the specified models. Additionally, please include the ICC for the random intercept only model. How did the model account for the high degree of missingness in some variables that were included as covariates such as race/ethnicity and parental education; what was the impact of the missingness on the final analytic sample included in the models? Please include some discussion on the compositional nature of the data given that both ST and TPA are analyzed, but in separate models.
Gupta, N., Mathiassen, S. E., Mateu-Figueras, G., Heiden, M., Hallman, D. M., Jorgensen, M. B., & Holtermann, A. (2018). A comparison of standard and compositional data analysis in studies addressing group differences in sedentary behavior and physical activity. The International Journal of Behavioral Nutrition and Physical Activity, 15(1), 53. https://doi.org/10.1186/s12966-018-0685-1
RESPONSE 1.4. A statement regarding the descriptive statistics has been added on Lines 141-142 to read “Participant characteristics were summarized using frequencies and percentages for categorical data”.
Information regarding the assumption tests undertaken have been added to Lines 147-153. The following sentences were added to the manuscript: “Linear regression analyses were undertaken assuming a linear relationship, multivariate normality, homoscedasticity and little multicollinearity which were tested via inspection of: scatter plots of the outcomes vs the independent variable; histograms of the outcome variables; scatter plots of the residual errors vs the linear predictor; and variance inflation factors of the variables included in the models respectively. Results from the assumption tests clarified that these assumptions had been met (data not shown).”
The ICCs have been calculated and have been added to Table 3 and Supplementary Table S2. An additional sentence has been added to Lines 153-154 to read as “Intraclass correlation coefficients (ICCs) and R-squared values (R2), as proposed by Snijders and Bosker[42], were calculated for each of the models".
In response to the reviewer’s query regarding missingness, we think there may be some confusion with the regression models. Each potential correlate was analysed in separate models, for each minutes spent in ST, TPA and MVPA as outcomes; we have made an amendment on Line 146 to make this clearer: “Adjusted multilevel linear regression models were used to determine the association between ST, TPA and MVPA for each potential correlate”. The models did not adjust for ethnicity or parental education (only age, gender, season, minutes of wear time and study clustering effects as stated on Lines 146-147) so this wouldn’t have an impact on the missingness on the final analytical sample for the other models. The missingness of ethnicity and parental education data has already been stated as a limitation in terms of having smaller sample sizes on Lines 312-314 but we thank the reviewer for pointing this out therefore we have added another limitation on Lines 340-342 to read “Estimates for the ethnicity and parental education variables had large amounts of missingness therefore we have assumed that these estimates would apply if the data was not missing”.
The compositional nature of the data has been added on Lines 359-361 to read “The data is compositional in nature, therefore using compositional data techniques as opposed to standard techniques may have produced different results[62]”.
REVIEWER 1.5. Line 162-177: Please include effect sizes for chi-squared tests
RESPONSE 1.5. Chi-squared test effect sizes (X2) have been added on Lines 182 and 184.
REVIEWER 1.6. Line 167-169: Are these statements qualitative comparisons, or were statistical tests run? Please clarify the statistical significance of the observed differences…
RESPONSE 1.6. No statistical tests were run, the statement was pointing out that Belgium had the lowest percentage of children meeting the guidelines compared to the USA which had the highest percentage. We thank the reviewer for suggesting to improve the readability so this sentence has been reworded to make it clearer on Lines 177-179 to read “The lowest percentage of children achieving guideline in MVPA levels was observed in Belgium (50.0%) and the highest percentage was observed in the USA (88.7%)”.
REVIEWER 1.7. Table 2: Please include effect sizes
RESPONSE 1.7. Chi-squared test effect sizes (X2) have been added to Table 2.
REVIEWER 1.8. Figure 1: Please consider presenting these figures as % of wear time on the y-axis to help clarify, especially, the relationship between ST and TPA. Also, a footer that describes each of the subplots will help to reinforce the meaning of each aspect of the multipaneled figure. ST is used throughout the text referring to sedentary time; however, the figures use SB
RESPONSE 1.8. To keep the three aspects of the paper consistent (compliance of the guidelines, patterns across the day and correlate regression analyses), we decided to present the minutes of wear time throughout, as opposed to presenting the percentage of wear time. We thank the reviewer for this suggestion, a footer has been added to the bottom of Figure 1 on Lines 215-218 to help describe the subplots: “Figure 1A shows the by country differences in minutes spent in ST, TPA and MVPA by hour. Figure 1B shows the differences in minutes spent in ST, TPA and MVPA by hour on weekdays compared to weekends. Figure 1C shows the differences in minutes spent in ST, TPA and MVPA by hour when the hours of daylight are less than 12 hours long compared to being more than 12 hours long.” Thank you for pointing out this typo, SB has been changed to ST in Figure 1.
REVIEWER 1.9. Line 183-185: Instead of only (B) and (C) to identify subplots in Figure 1, please consider something like Figure 1B and Figure 1C to more easily identify what is meant by the letters main text body.
RESPONSE 1.9. Thank you for this suggestion, this has been amended on Lines 196, 199 and 204.
REVIEWER 1.10. Lines 205- 217: What is the Adjusted R squared value for each of the respective models? Additionally, some of the covariates (e.g., season, time of sunrise, time of sunset, hours of daylight) seem like they might be highly correlated (i.e., may have shared variances) was there a search for optimal model fit when including these covariates? What were the considerations for multicollinearity between covariates in specifying the final model?
RESPONSE 1.10. The R-squared values have been calculated and added to Table 3 and Supplementary Table 2. An additional sentence has been added to Lines 153-154 to read as “Intraclass correlation coefficients (ICCs) and R-squared values (R2), as proposed by Snijders and Bosker[42], were calculated for each of the models”.
As mentioned in RESPONSE 1.4, we think there may be some confusion with interpretation of the regression model output in Table 3, therefore the regression analyses have been clarified on Line 146: “Adjusted multilevel linear regression models were used to determine the association between ST, TPA and MVPA for each potential correlate”. Each row represents the regression output for each single potential correlate with either minutes spent in ST, TPA or MVPA as the outcome. Each model was adjusted for age, gender, season, minutes of wear time and study clustering effects (unless age, gender or season were the potential correlate of interest), so concerns about multicollinearity only apply to these covariates plus each hypothesised correlate. Multicollinearity was assessed and reported to address the reviewer’s previous comment (REVIEWER 1.4) so please see RESPONSE 1.4.
REVIEWER 1.11. Line 258: ‘were’ instead of ‘was’
RESPONSE 1.11. This has been changed in the manuscript on Line 278.
REVIEWER 1.12. Lines 263-281: It remains unclear what statistical analyses were conducted to arrive at these discussion points, or if this is based upon qualitative description of the plots alone. Please clarify this point. It is stated that ST decreases, but the plots show that first ST increases, then it decreases after reaching a fairly clear peak point or else reaching several local maxima throughout the day...
RESPONSE 1.12. This has been clarified on Line 284 and the description of the ST levels have been amended to better reflect the patterns across the day on Lines 194-196 “Figure 1 shows that ST levels increase until around 09:00 and then decrease throughout the day whereas TPA and MVPA levels increase throughout the day with variations in PA by country, day of the week and hours of daylight between 11:00-15:00” and 284-285 “Visual inspection of the plots of ST by hour suggested that children spent fewer minutes in ST as the day progressed after an initial increase in ST levels until 09:00”.
REVIEWER 1.13. Lines 329-331: Did you mean to say something about “sleep/nap” time here instead of “non-wear time”?
RESPONSE 1.13. Sleep/nap time has been added on Line 358 to make this sentence clearer.

Reviewer 2 Report
Thank you for inviting me to review this article entitled, “International comparison of the levels and potential correlates of objectively measured sedentary time and physical activity among 3-4 year old children.” Because different research teams use different accelerometer cut points when processing data across studies, it is difficult to understand the extent to which preschool children meet daily recommendations for physical activity (Beets, Bornsetin, Dowda, & Pate, 2011). As such, the current study seeks to address this problem by using a consistent approach to process accelerometry data from >1,000 young children, who participated in six studies in four countries. Additionally, the authors sought to identify correlates of preschoolers’ physical activity and sedentary time using demographic and environmental variables that were available in the International Children’s Accelerometry Database. As the authors point out, this is important because the identification of mediators and moderators are key to the development of behavior change interventions. Overall, I believe this study was well conceptualized and executed, and I am supportive of its publication in the International Journal of Environmental Research and Public Health. The following comments are intended to strengthen this study:
1. A main goal of this study was to make statements about the percent of children meeting PA recommendations in four countries (US, UK, Belgium, and Switzerland), and to compare young children’s sedentary time and physical activity levels across these countries. I am concerned about the relatively small number of children in each of these countries that was included in the sample and the extent to which the children were representative of each country’s population. For example, there were only 380 children in the US sample, and they all came from the Columbia, South Carolina area. Similarly, there were only 104 children in the Belgian sample. I did not see the authors recognize this as a limitation in the study. Also, the authors should be careful when making statements that generalize their findings to the country that the children in the sample came from (e.g., p. 1, lines 34-37; p. 5, lines 168-169).
2. The authors pointed out the importance of their correlational analyses for the development of interventions that are designed to increase young children’s physical activity levels. I was hoping that they would have put their findings into this context more explicitly by providing guidance or recommendations for intervention developers. If they can expand upon this in the discussion section, it would be helpful.
Reference:
Beets, M. W., Bornstein, D., Dowda, M., & Pate, R. R. (2011). Compliance with National Guidelines for Physical Activity in U.S. Preschoolers: Measurement and Interpretation, Pediatrics, 127 (4):658-64
Author Response
17th May 2019
Dear Sir/Madam,
We appreciate the opportunity to resubmit our manuscript “International comparison of the levels and potential correlates of objectively measured sedentary time and physical activity among 3-4-year old children”, ID: ijerph-496064, for publication as an original article in the International Journal of Environmental Research and Public Health.
The comments from the reviewers helped us improve our manuscript. Our response to each comment is indicated in red text in the following pages. Revisions in the manuscript are shown as tracked changes. We hope that the responses and revisions are sufficient to make our manuscript suitable for publication in your journal.
We look forward to hearing from you at your earliest convenience.
Yours sincerely,
The study authors
Response to Reviewer 2 Comments
REVIEWER 2.1. A main goal of this study was to make statements about the percent of children meeting PA recommendations in four countries (US, UK, Belgium, and Switzerland), and to compare young children’s sedentary time and physical activity levels across these countries. I am concerned about the relatively small number of children in each of these countries that was included in the sample and the extent to which the children were representative of each country’s population. For example, there were only 380 children in the US sample, and they all came from the Columbia, South Carolina area. Similarly, there were only 104 children in the Belgian sample. I did not see the authors recognize this as a limitation in the study. Also, the authors should be careful when making statements that generalize their findings to the country that the children in the sample came from (e.g., p. 1, lines 34-37; p. 5, lines 168-169).
RESPONSE 2.1. We thank the reviewer for pointing out this limitation, this has been added to the discussion section on Lines 347-350 to read “It is important to acknowledge that there are a relatively small number of children in each of the countries that were included in the sample, therefore our findings are not representative of each country’s population”.
Statements referring to country have been amended to tone down the generalisability on Lines 35 “Across the UK, Switzerland, Belgium and the USA, children in our analysis sample spent 490 minutes in ST per day and 30.0% and 21.2% of children did not engage in recommended daily TPA (≥180 minutes) and MVPA (≥60 minutes) guidelines” and Lines 176-177 “Our findings suggest that the percentage of children reaching TPA and MVPA guidelines varied between the different countries”.
REVIEWER 2.2. The authors pointed out the importance of their correlational analyses for the development of interventions that are designed to increase young children’s physical activity levels. I was hoping that they would have put their findings into this context more explicitly by providing guidance or recommendations for intervention developers. If they can expand upon this in the discussion section, it would be helpful.
RESPONSE 2.2. We thank the reviewer for the comment, but it is not possible to recommend specific intervention content based on the findings from this study. We therefore think that the current text on Lines 368-381 which summarises the study findings is appropriate and that any further changes would inevitable be speculative and go beyond what is possible with the study findings. The limitations surrounding correlational analyses has been discussed as a limitation on Lines 363-365: “Data from longitudinal studies can estimate modifiable factors associated with changes in ST and PA[63], whereas our cross-sectional study is limited in only providing evidence of associations”.

Reviewer 3 Report
This is perhaps the best written journal article I have ever read as an initial submission for peer review. I felt like I was reading a final article, not an initial submission. It is clear the authors spent a considerable amount of time and effort in compiling and editing this draft, and they should be commended.
The study is also very well done as such and will add to the literature in the of preschool children’s sedentary time and physical activity. I appreciate that the authors analyzed ST, TPA, and MVPA separately rather than on a continuum, as they each are different constructs with different influences and correlates. Notable strengths are the use of a centralized database using the same accelerometer and a standardized method of processing the data. Another strength is the use of timeanddate.com to assess the association of daylight hours, sunrise, and sunset times with sedentary and physical activity in a large sample of children from different countries in the northern hemisphere. This is a great use of two databases to answer an important clinical and research question, and with sufficient power, and thus moves the field forward significantly. Figure 1 was particularly instructive and valuable.
The limitation that the data are over 10 years old is noted in the limitations section and is more than compensated by the study’s considerable strengths.
A couple of very minor editorial suggestions:
Figure 1 lists the abbreviation SB, presumably for sedentary behavior, rather than ST, which is used in the article.
Page 11 lines 235-248 were a bit difficult to read/interpret, I think because of the mental gymnastics necessary to keep track of the order of prevalence of compliance with guidelines, Countries, and thresholds. I would recommend separating out those countries that had higher prevalence rates (UK and Canada) from those who had lower prevalence rates (Belgium and Australian), and then it might be easier to link this to the threshold levels for TPA. Actually, since the data from Canada is counter to point made by UK, Belgium, and Australia, I would recommend not listing Canada until later in the paragraph when you the point about differences in accelerometry wear time. Can you also explicitly state the difference in wear time?
Author Response
17th May 2019
Dear Sir/Madam,
We appreciate the opportunity to resubmit our manuscript “International comparison of the levels and potential correlates of objectively measured sedentary time and physical activity among 3-4-year old children”, ID: ijerph-496064, for publication as an original article in the International Journal of Environmental Research and Public Health.
The comments from the reviewers helped us improve our manuscript. Our response to each comment is indicated in red text in the following pages. Revisions in the manuscript are shown as tracked changes. We hope that the responses and revisions are sufficient to make our manuscript suitable for publication in your journal.
We look forward to hearing from you at your earliest convenience.
Yours sincerely,
The study authors
Response to Reviewer 3 Comments
REVIEWER 3.1. Figure 1 lists the abbreviation SB, presumably for sedentary behavior, rather than ST, which is used in the article.
RESPONSE 3.1. Thank you for pointing out this typo, SB has been changed to ST in Figure 1.
REVIEWER 3.2. Page 11 lines 235-248 were a bit difficult to read/interpret, I think because of the mental gymnastics necessary to keep track of the order of prevalence of compliance with guidelines, Countries, and thresholds. I would recommend separating out those countries that had higher prevalence rates (UK and Canada) from those who had lower prevalence rates (Belgium and Australian), and then it might be easier to link this to the threshold levels for TPA. Actually, since the data from Canada is counter to point made by UK, Belgium, and Australia, I would recommend not listing Canada until later in the paragraph when you the point about differences in accelerometry wear time. Can you also explicitly state the difference in wear time?
RESPONSE 3.2. We thank the reviewer for suggesting reordering the sentences in this paragraph to help improve the readability. The sentences have been swapped around and reworded so that the TPA thresholds are discussed before the MVPA guidelines on Lines 265-270. The Canadian reference doesn’t provide the minutes of wear time data so we are unable to state this.

Round 2
Reviewer 1 Report
Thank you for your thorough responses!